# Oral Isavuconazole Combined with Nebulized Inhalation and Bronchoscopic Administration of Amphotericin B for the Treatment of Pulmonary Mucormycosis: A Case Report and Literature Review

**DOI:** 10.3390/jof10060388

**Published:** 2024-05-29

**Authors:** Xuan Leng, Hui Zhou, Zhiyang Xu, Feng Xu

**Affiliations:** 1Department of Infectious Diseases, The Second Affiliated Hospital, Zhejiang University School of Medicine, Hangzhou 310009, China; 3190102430@zju.edu.cn (X.L.); huizhou93@zju.edu.cn (H.Z.); 2515164@zju.edu.cn (Z.X.); 2Research Center for Life Science and Human Health, Binjiang Institute, Zhejiang University, Hangzhou 310053, China; 3Key Laboratory of Novel Targets and Drug Study for Neural Repair of Zhejiang Province, School of Medicine, Hangzhou City University, Hangzhou 310015, China

**Keywords:** *Rhizopus microsporus*, pulmonary mucormycosis, amphotericin B, isavuconazole, immunocompromised patient, case report

## Abstract

Pulmonary mucormycosis (PM) is an invasive and potentially fatal fungal infection, with *Rhizopus microsporus* (*R. microsporus*) being the most common pathogen. The routine therapy for this infection includes surgery and antifungal agents. However, the therapeutic effects of single agents are unsatisfactory due to the rapid progression of mucormycosis, while not all patients can tolerate surgery. Innovative treatment methods like combination therapy await validations of their clinical efficacy. We report a case of PM that was diagnosed via metagenomics next-generation sequencing (mNGS) of black drainage fluid from the patient’s lung. The patient eventually recovered and was discharged after a combination therapy of oral isavuconazole, inhaled amphotericin B, and local perfusion of amphotericin B through bronchoscopy, which may be a promising strategy for the treatment of PM, especially for cases where surgery is not possible. A retrospective study of 297 cases in a literature review highlights the different treatment methods used in clinical practice.

## 1. Introduction

Mucormycosis is an opportunistic and invasive fungal infection caused by members of the order Mucorales, among which *Rhizopus*, *Mucor*, and *Rhizomucor* are the most common. Other pathogenic organisms include *Lichtheimia* (*Absidia*), *Cunninghamella*, *Syncephalastrum*, and *Saksenaea* [1]. The predisposing factors for mucormycosis include uncontrolled diabetes mellitus (especially with ketoacidosis), hematological malignancy, stem cell or solid organ transplant, and corticosteroid use. Mucormycosis has also recently been observed in COVID-19 patients [2,3]. The most affected body sites are the orbital-cerebral regions, lung, and skin, among which pulmonary mucormycosis (PM) accounts for 24–30% of all cases [4,5]. While orbital-cerebral infection is often seen in people with diabetes mellitus, PM occurs frequently in immunocompromised individuals [6]. The clinical manifestation is not specific and often comprises chest pain, cough, hemoptysis, and fever. Such symptoms are also found in association with arterial aneurysms, pseudoaneurysms, and isolated cavernous lesions [7]. There have also been reports of PM mimicking a lung tumor [8,9]. The infection tends to be fatal without timely diagnosis and treatment and its mortality rate is reported to be over 50% [10,11,12].

Over recent years, multidisciplinary treatment approaches involving antifungal medicine and aggressive surgical debridement have been recommended [13]. However, some patients may not be able to tolerate surgical treatment due to their physical condition. The currently available antifungal drugs contain amphotericin B (AmB), posaconazole, and isavuconazole. Nonetheless, the penetration of any single antifungal medicine is poor and often accompanied by serious systemic reactions, such as a rash or kidney damage [14]. The required dosage and duration of antifungal agents remain unclear. Moreover, there has been no definitive evidence to clarify the effectiveness of combination therapy, and the efficacy of other adjunctive therapies, such as hyperbaric oxygen and drug atomization, is unclear [15]. Therefore, more clinical studies are needed to explore potential effective options for treatment.

Here, we report a case of PM with symptoms of fever, chest tightness, and hemoptysis. The patient recovered well after receiving a combination therapy of oral posaconazole with AmB via inhalation and local airway perfusion. This case highlights the importance of combination therapy with an oral antifungal agent and adjunctive therapies (nebulized inhalation and bronchoscopic administration of an antifungal agent). We hope that the combination treatment used in this case will be a promising strategy for the treatment of PM.

## 2. Case Report

A 70-year-old man presented with right hip pain after falling at home. The X-ray image suggested a right femoral neck fracture. He had a medical history of hypertension (20 years) and acute myeloid leukemia (AML) 4 years prior (treated with azacitidine). It is worth mentioning that he was hospitalized for COVID-19 infection in another hospital for one month and was discharged one day prior. He denied any history of smoking or alcohol, surgical trauma, allergies, travel contact, or related family illnesses. The Department of Orthopedics planned to perform surgical treatment. On the first day after admission, chest computed tomography (CT) indicated a thick-walled cavity in the left lung for which differentiation was required between an abscess with cavity formation and a tumor. The patient was thus transferred to the Department of Infectious Disease due to elevated inflammatory markers and a sustained increase in body temperature.

After the transfer, the patient’s body temperature rose and was maintained at 38–39 °C. He denied having any cough, sputum, dyspnea hemoptysis, or chest tightness, but he showed obvious fatigue with the raised temperature. Laboratory testing results showed pancytopenia with a white cell count of 3200/μL (neutrophil 2810/μL), a hemoglobin level of 6.5 g/dL, and a platelet (PLT) count of 31,000/μL. The lymphocyte level was also reduced, with a count of 260/μL. The erythrocyte sedimentation rate (ESR) was elevated, at 98 mm/h. The biochemical routine test showed impaired liver function with a direct bilirubin (DBil) level of 7.1 μmol/L, an albumin/globulin (A/G) ratio of 0.90, a total protein amount of 50.9 g/L, and an albumin level of 24.1 g/L. The C-reactive protein (CRP) level was 255 mg/L. The level of procalcitonin was 1.34 ng/mL. Two tumor markers were increased, with a cellular keratin fragment level of 8.6 ng/mL and a squamous epithelial cell carcinoma-associated antigen level of 2.0 ng/mL. The test for T cell CD molecules showed a reduced lymphocyte percentage (CD45+) (3.99%). The other laboratory test results were normal. Re-examination via chest CT at day 6 after admission revealed a thick-walled cavity in the lower lobe of the left lung that was enlarged compared to that in the previous film (day 1) and accompanied by obstructive pneumonia. Combining the image, laboratory test results, and high fever, we considered an abscess. The patient was treated with intravenous meropenem (0.5 g q8h) immediately after sputum and blood samples were sent for cultivation. However, the patient’s body temperature remained high after five days of antibiotic treatment. The levels of white cells and neutrophils continued to drop to 2000/μL and 1730/μL, respectively. 

An intravenous treatment with voriconazole (0.2 g q12h) was added at first because a sputum test at the previous hospital was positive for *Aspergillus*. Nevertheless, the patient still had a persistent high fever after 7 days of voriconazole treatment. No significant decrease in inflammatory indicators or absorption of the left lung abscess was observed. In addition, the patient developed a cough, and occasional blood streaks were visible in their sputum. Puncture drainage was performed on the abscess in the lungs, resulting in the discharge of black fluid (Figure 1). Considering the patient’s immunodeficiency and infection history of COVID-19, the possibility of another fungal infection still could not be ruled out. Since the sputum smear and blood culture did not show any positive outcomes, we sent the drainage fluid for metagenomics next-generation sequencing (mNGS) and cultivation given the rapid progress of fungal infections. The cultivation test showed negative results. The mNGS detection of pathogenic microorganisms in the hydrothorax showed *Rhizopus* (genus: *Rhizopus*, relative abundance: 89.8%; species: *Rhizopus microsporus* (*R. microsporus*), number of sequences: 46; specificity: 99%). A surgical consultation did not suggest surgical indications. Instead, AmB was administered intravenously at a dose of 5 mg/d and gradually increased to maintain a dose of 50 mg/d for treatment. Simultaneously, we performed a CT-guided puncture for pathological examination and excluded the possibility of a tumor as the result showed necrotic and degenerative tissue. A bone marrow smear examination also helped rule out an acute leukemia attack. A sputum culture showed methicillin-resistant *Staphylococcus aureus* (MRSA), and an anti-infective treatment was administered with intravenous linezolid (600 mg q12h) and oral moxifloxacin (400 mg qd).

After 22 days of AmB treatment (accumulated dose, 830 mg), the patient’s body temperature dropped to around 37 °C without any rebound. However, there were repeated occurrences of low potassium during use, which was considered a serious side effect of the AmB. Oral isavuconazole (200 mg qd) was administered instead, and the linezolid was stopped due to the decreased PLT. The patient’s body temperature stabilized, and their CRP level decreased to around 80 mg/L. 

After two weeks of isavuconazole treatment, a chest CT scan showed absorption of the lesion, but pleural effusion appeared (day 44). Moreover, the patient presented with an obvious cough, sputum production, and chest tightness accompanied by hemoptysis, with the CRP level rebounding to around 200 mg/L. Considering that angioinvasion is a pivotal feature of mucormycosis, nebulized inhalation of AmB (15 mg/d) was added to the treatment regimen, and perfusion of AmB (10 mg once) in the left lower lobe basal segment through bronchoscopy was given two times (day 48, day 76). At the same time, auxiliary treatments of targeted blood transfusion, such as terbinate (recombinant human thrombopoietin) and recombinant human granulocyte stimulating factor (rhGM-CSF), were given to improve the low sustained blood triad, and thymalfasin was used to enhance the patient’s immune system. A chest CT scan (day 101) showed a thick-walled cavity in the lower lobe of the left lung with inflammatory drainage, with improved absorption compared to before. There was slight inflammation in both lungs, and absorption was observed compared to before. Cough and expectoration were significantly relieved, and hemoptysis did not occur. The patient’s body temperature was maintained at around 37 °C. The blood routine results showed that the levels of white blood cells, neutrophils, and platelets had returned to normal values. The biochemical routine showed that CRP was 34.7 mg/L, which was a significant reduction. The patient was discharged on day 103 and instructed to continue taking 200 mg qd of isavuconazole orally. One month later, the patient came again for a follow-up examination, which showed that the inflammatory indicators had decreased to normal. He also successfully underwent orthopedic surgery. The process of diagnosis and treatment is presented in Figure 1 and Figure 2.

## 3. Research Methods 

We carried out a systematic literature search in PubMed with the following search terms limited to the title or abstract: ((Mucormycosis) OR (Zygomycosis) OR (Mucormycose) OR (Rhizopus) OR (Lichtheimia) OR (Mucor) OR (Rhizomucor) OR (Cunninghamella) OR (Syncephalastrum) OR (Saksenaea) OR (Apophysomyces) OR (Actinomucor) OR (Thamnostylum) OR (Cokeromyces) OR (Mortierella)) AND ((pulmonary) OR (lung)) [16]. Only case reports published in English between 1 January 2010 and 30 November 2023, satisfying the 2018 EORTC/MSG criteria, were included [17]. In total, 974 articles were found. Two authors (LX and ZH) independently collected these articles and evaluated their appropriateness based on the title and abstract with regard to the 2018 EORTC/MSG criteria. Finally, 262 articles including 297 patients were selected. The details are described in Appendix A.

## 4. Discussion

Among the 297 patients selected, the average age was 48 years, and 68% of them were male. The most common cause of PM was *Rhizopus*, with a proportion of 46.5% (79/170). The most common comorbidity among the patients was diabetes mellitus in 109 (109/297, 36.7%) patients, followed by hematological malignancy (87/297, 29.3%) and stem cell (13/297, 4.4%) or organ transplant (29/297, 9.8%), consistent with a previous study [18]. The most common clinical presentations comprised fever (44%), chest pain (37%), hemoptysis (16.8%), and dyspnea (14%). It is noteworthy that four patients presented with hoarseness of voice (a mass compressing the vagus nerve) and two patients presented with Pancoast syndrome, which means that PM could mimic lung cancer [19]. Consolidation (79/297, 26.6%), a cavity (19/297, 6.4%), infiltration (37/297, 12.5%), pleural effusion (41/297, 13.8%), and a reversed halo sign (10/297, 3.4%) were usually observed in chest CT images but were not distinctive. Pseudoaneurysm, embolism, and lymphadenopathy were also noted [20]. The trachea, mediastinum, and ribs could also be affected by PM. The majority of the cases were diagnosed through culture (78/297, 26.3%) and histopathology (122/297, 41.1%), with samples obtained from bronchoalveolar lavage fluid (BALF), biopsy, and autopsy. With recent advancements, molecular identification techniques such as polymerase chain reaction (PCR) (26/297, 8.8%) and mNGS (12/297, 4.0%) have also been widely utilized for early diagnosis and classification.

Regarding the therapy, the current mainstream approach is treatment with a single antifungal agent, among which AmB (185/270, 68.5%) is applied first, followed by posaconazole (66/270, 24.4%) and isavuconazole (20/270, 7.4%). Antifungal agents combined with surgery are also commonplace (75/270, 27.8%). Combination therapy accounts for 12.2% (33/270). The lowest death rate of 24% (18/75) was observed in a group receiving antifungal agents plus surgery. Compared with that under single antifungal agents (71/148, 48.0%), the mortality under combination therapy was reduced (13/33, 39.4%) [21]. Clinical information about the 297 patients was listed in Table 1. We can therefore summarize that combination therapies of surgery and antifungal agents may be an effective approach. However, immunodeficient patients comprise a large proportion of those with PM, which means that surgery may not be an option. In this situation, combination therapies and other adjunctive therapies may be significant. It is notable that seven patients treated with nebulized AmB obtained a favorable prognosis [22,23,24,25,26,27,28]. Among them, one case reported the successful treatment of PM through the intravenous inhalation and local airway perfusion of AmB [28]. However, it is worth mentioning that this patient was a 19-year-old girl who had intact immunity and did not show any side reaction to the AmB. Therefore, our case demonstrated that the use of oral isavuconazole combined with the nebulized inhalation and bronchoscopic administration of AmB may be promising for immunodeficient patients with PM. 

The incidence of opportunistic mucormycosis, especially PM, has been increasing during recent years, probably owing to the longer survival of immunocompromised patients and progress in examination techniques [29]. Susceptibility factors play a significant role in mucormycosis infection. Conditions involving immunodeficiency are the most common factor, among which diabetes mellitus is considered the most significant [30]. Other risk factors, such as stem cell or solid organ transplant, hematological malignancy, and autoimmune diseases, cannot be ignored. The patient in our case had a history of AML just a half year prior to admission. Besides this, he experienced a COVID-19 infection and was treated with steroids at another hospital for over one month not long prior. Thus, the prompt rectification of predisposing conditions is indispensable and urgent. Thymalfasin was also applied to enhance the patient’s immunity, and immune adjuvant therapies were applied, including granulocyte macrophage colony-stimulating factor (GM-CSF), checkpoint blockage, interferon gamma (IFN-γ), and interleukin (IL)-7 [31,32]. In a recent report, IL-7, GM-CSF, and G-CSF were administered together with L-AmB and posaconazole to successfully cure a lymphopenic patient with aplastic anemia and mucormycosis [33].

Rapid progress is a major characteristic of PM. Therefore, early diagnosis makes a significant difference in achieving a better prognosis and decreasing the need for surgery [34,35]. Generally, the diagnosis of mucormycosis relies on the availability of radiography-, histopathology-, culture-, and molecular-based methods [4]. The radiological findings of PM often suggest infiltrates, effusion, cavities, and masses, which conventionally present on the upper lobes [20]. Intrapulmonary masses with a halo sign (central consolidation with surrounding ground-glass opacity) on CT images may be suggestive of an invasive fungal infection, particularly in immunocompromised patients [36], but they are not specific to mucormycosis. The ‘T2 hypointense rim sign’ in MRI is also described in several cases and may be diagnostically significant [37]. In our case, the patient’s CT image displayed a pulmonary abscess caused by infection, and we promptly performed CT-guided puncture drainage, from which we collected a black liquid (Figure 1). The black drainage and the restriction of immune function led us to suspect a fungal infection. Traditional methods like histopathology and culturing are time-consuming and lack specificity [38]. Molecular-based methods can be a good alternative and even an improvement, especially when the culture is negative or the availability of materials is limited [39,40]. Ribosomal DNA sequences including the V9 region of the 18S ribosomal RNA, the D1/D2 domains of the 28S ribosomal DNA, and the ITS region are also useful for identification [41]. mNGS is a rapid and non-invasive diagnostic method that has an important function when a rare pathogen is suspected. Additionally, multiple diagnostic techniques including percutaneous needle biopsy, pleural fluid drainage, and bronchoalveolar lavage specimens from fiber optic bronchoscopy can provide sufficient samples for histopathological and molecular detection and diagnosis [42]. In this case, mNGS sensitively detected *Mucorales* in the lung, and early anti-mucor treatment was started immediately.

Like prompt diagnosis, proper antifungal therapy is also tightly associated with the prognosis. The first recommendations for the diagnosis and treatment of mucormycosis were proposed after the 3rd European Conference on Infections in Leukemia (ECIL-3). They pointed to lipid formulations of AmB as the drug of choice with lower toxicity [43]. Their proposal also suggested a combination therapy of AmB, echinocandins, and GM-CSF. In 2017, ECIL-6 reported that AmB, posaconazole, and isavuconazole are the most potent agents in vitro [44]. The latest global guidelines issued by the European Confederation of Medical Mycology (ECMM) recommend aggressive surgery combined with antifungal agents [4]. Given the poor penetration of antifungal agents, prompt surgery may prevent dissemination and invasion into blood vessels, particularly in patients with lesions limited to only one lung lobe [14]. Choi H et al. reported that surgical resections, such as lobectomy, pneumonectomy, and wedge resection, improve the survival rate of PM [45]. These mainstream therapies are summarized in detail in the literature review above.

In our case, the patient could not receive surgery due to their pancytopenia and compromised immune function. We could only select antifungal medicines. AmB is strongly recommended as the mainstay treatment in cases with any organ involvement [4,46]. Thus, we first gave intravenous AmB 5 mg daily, gradually increasing to 50 mg daily. However, after 22 days of administration, the patient’s temperature remained high, and they experienced one episode of systemic rash (allergic reaction) and repeated hypokalemia, which we considered to be side effects of AmB. In view of this condition, we swapped AmB for oral isavuconazole (200 mg daily). Isavuconazole is a broad-spectrum triazole antifungal agent; its target is the CYP51A enzyme, which is indispensable for ergosterol synthesis and cell membrane formation [47]. It has been nominated as a first-line treatment for mucormycosis with equivalent efficacy to AmB and is an appropriate alternative when a patient cannot tolerate AmB toxicity [4]. However, the patient’s inflammatory indicators were not improved, and one episode of hemoptysis occurred after nearly one month of isavuconazole administration. Considering the angioinvasion ability of fungus, AmB inhalation was added to the treatment regimen, and we also performed two perfusions of AmB to the left lower lobe basal segment under bronchoscopy (days 48 and 76). In addition to intravenous AmB, AmB inhalation has been utilized in the clinic to provide higher concentrations in local regions and reduced systematic side effects [48]. During the treatment, the patient’s potassium levels returned to normal, and no liver or kidney damage was observed. However, there is a paucity of clinical clues about the usage of AmB through bronchoscopy. In recent years, there has indeed been a case report on the successful treatment of *R. microsporus* pulmonary infection in an immunocompetent patient with AmB administered via inhalation and bronchoscopy [28]. However, until now, there has been no definitive evidence to clarify the effectiveness of combination therapy. An animal experiment demonstrated that AmB combined with micafungin or anidulafungin is more effective than AmB alone in treating murine mucormycosis [49]. A prospective study in Russia also proved that AmB combined with echinocandins can improve the overall survival rate, with reduced side effects [50]. In any case, the use of combination therapy is rational given its limited toxicity and potential but unproven benefit. In our case report, the immunocompromised patient progressively improved with the combination of isavuconazole, nebulized AmB, and two rounds of bronchoscopic therapy. This proves the efficacy and low side effects of nebulized AmB inhalation and bronchoscopic AmB administration, which may be a promising method for the treatment of PM. More clinical studies are essential to validate the efficacy and safety of this combination therapy.

## 5. Conclusions

PM is an acute, opportunistic infection that can be fatal unless it is diagnosed early. Improvements in the survival rate depend on the prompt rectification of predisposing conditions, early diagnosis, and effective treatments, including systemic antifungal therapy and surgical resection. This case highlights that a combination therapy of oral posaconazole, AmB inhalation, and bronchoscopic perfusion of AmB may be a promising method to treat PM.

## Figures and Tables

**Figure 1 jof-10-00388-f001:**
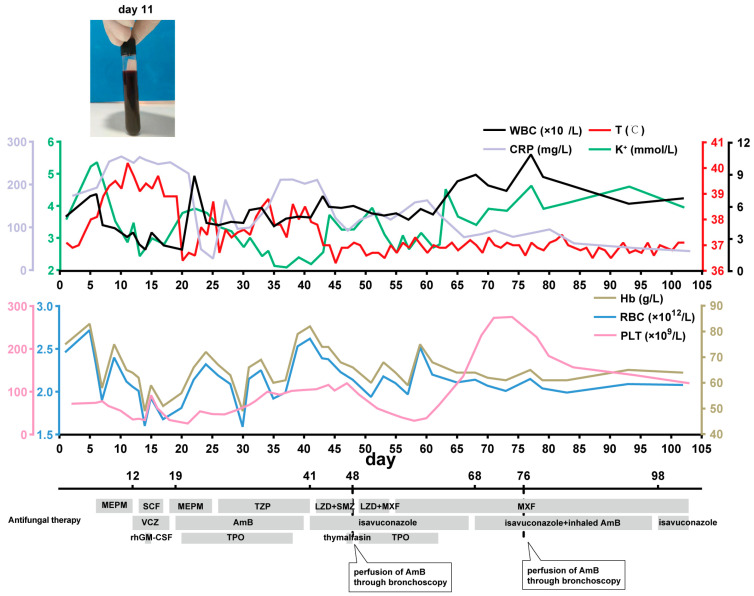
The process of treatment and changes in relevant indicators during hospitalization. Description: WBC: white blood cell; T: temperature; K: potassium; CRP: C-reactive protein; Hb: hemoglobin; RBC: red blood cell; PLT: platelet; MEPM: meropenem; SCF: cefoperazone sulbactam; TZP: piperacillin tazobactam; LZD: linezolid; SMZ: sulfamethoxazole; MXF: moxifloxacin; VCZ: voriconazole; AmB: amphotericin B; rhGM-CSF: recombinant human granulocyte-macrophage colony stimulating factor; TPO: thrombopoietin.

**Figure 2 jof-10-00388-f002:**
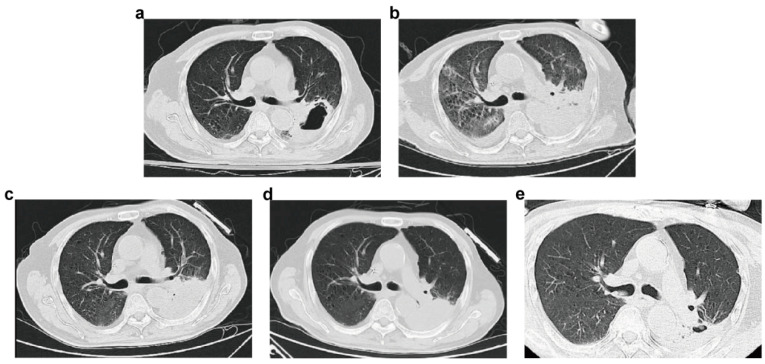
Changes in pulmonary lesions in chest CT of the patient. (**a**): Chest CT image of the patient at day 1 after admission. (**b**): Chest CT image at day 44 after admission. (**c**): Chest CT image at day 70 after admission. (**d**): Chest CT image at day 101 after admission. (**e**): Chest CT image at day 69 after discharge.

**Table 1 jof-10-00388-t001:** Demography, pathogens, clinical manifestations, risk factors, sites of involvement, radiography, diagnosis, therapy and prognosis of 297 PM patients reported in the literature.

Parameters				N = 297	Death
	Age in years			48 (0.58–88)	
	Male/Female			200/95	
Pathogen genus					
	*Actinomucor*			1/297 (0.3%)	
	*Cunninghamella*			25/297 (8.4%)	
	*Cokeromyces*			2/297 (0.7%)	
	*Mucor*			33/297 (11.1%)	
	*Rhizopus*			79/297 (26.6%)	
	*Rhizomucor*			15/297 (5.0%)	
	*Syncephalastrum*			2/297 (0.7%)	
	*Lichtheimia*			13/297 (4.4%)	
	Untyped			127/297 (42.8%)	
Clinical manifestation					
	Cough			111/297 (37.4%)	
	Fever			132/297 (44.4%)	
	Chest pain			111/297 (37.4%)	
	Hemoptysis			50/297 (16.8%)	
	Dyspnoea			42/297 (14.1%)	
	Hoarseness of voice			4/297 (1.3%)	
	Pancoast syndrome			2/297 (0.7%)	
Risk factors					
	Diabetes mellitus			109/297 (36.7%)	
	Hematological malignancy			87/297 (29.3%)	
	Stem cell or solid organ transplant			42/297 (14.1%)	
		Renal transplant		22	
	Use of glucocorticoids			29/297 (9.8%)	
	Immunosuppressive therapy			28/297 (9.4%)	
	Autoimmune diseases			18/297 (6.1%)	
	COVID-19			40/297 (13.5%)	
	Tuberculosis			18/297 (6.1%)	
	Chronic kidney disease			25/297 (8.4%)	
	Cancers			10/297 (3.4%)	
	HIV infection			5/297 (1.7%)	
	Contact history			9/297 (3.0%)	
	Immunocompetent			26/297 (8.8%)	
CT findings					
	Consolidation			79/297 (26.6%)	
	Pleural effusion			41/297 (13.8%)	
	Infiltration			37/297 (12.5%)	
	Mass			31/297 (10.4%)	
	Cavity			19/297 (6.4%)	
	Reversed halo sign			10/297 (3.4%)	
	Pseudoaneurysm			7/297 (2.4%)	
	Lymphadenopathy			6/297 (2.0%)	
Site of involvement					
	Both			62/243 (25.5%)	
	Right			108/243 (44.4%)	
		RUL		26	
		RML		10	
		RLL		47	
	Left			71/243 (29.2%)	
		LUL		21	
		LMZ		4	
		LLL		24	
	No mention of the location of lobe			54	
	Trachea			6/297 (2.0%)	
	Mediastinum			18/297 (6.1%)	
	Rib destruction			3/297 (1.0%)	
Diagnosis method					
	Autopsy			21/297 (7.1%)	
	BALF			66/297 (22.2%)	
	Biopsy			231/297 (77.8%)	
		Culture		78	
		Histopathology		122	
	Molecular identification			61/297 (20.5%)	
		PCR		26	
		mNGS		12	
		DNA sequencing		1	
		ITS		13	
		MALDI-TOF		3	
Treatment strategy					
	No mention			27	
	Single antifungal agent			148/270 (54.8%)	71/148 (48.0%)
		Amphotericin B		185	1
			Inhaled AmB	2	0
			Inhaled + IV AmB	5	0
		Isavuconazole		20	6
		Itraconazole		3	0
		Posaconazole		66	15
		Voriconazole		2	1
		Fluconazole		1	0
	Antifungal agents (combination)			33/270 (12.2%)	13/33 (39.4%)
		AmB + posaconazole		11	2
			Inhaled AmB + posaconazole	2	0
	Sugery			14/270 (5.2%)	4/14 (28.6%)
	Antifungal agents + surgery			75/270 (27.8%)	18/75 (24.0%)
	Brochoscopy therapy of perfusion with AmB			1	0
Drug side effects					
	Amphotericin B				
		Nephrotoxicity		29	
		Electrolyte imbalance (hypokalemia)		6	
		Allergic reaction		1	
	Posaconazole				
		Gastrointestinal side-effects		1	
Treatment time (days)				138 (3–1140)	
Outcome					
	Died			130/285 (45.6%)	130
	Alive			155/285 (54.4%)	

COVID-19, coronavirus disease 2019; HIV, human immunodeficiency virus; RUL, right upper lobe; RML, right middle lobe; RLL, right lower lobe; LUL, left upper lobe; LMZ, left middle zone; LLL, left lower lobe; BALF, bronchoalveolar lavage fluid; PCR, polymerase chain reaction; mNGS, metagenomic next-generation sequencing; ITS, internally transcribed spacer; MALDI-TOF, matrix-assisted laser desorption ionization time-of-flight mass spectrometry; IV, intravenous.

## Data Availability

Data are contained within the article and Appendix A.

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
