# Peer review of "Oral Isavuconazole Combined with Nebulized Inhalation and Bronchoscopic Administration of Amphotericin B for the Treatment of Pulmonary Mucormycosis: A Case Report and Literature Review"

_jof, 2024, doi:10.3390/jof10060388_

Round 1
Reviewer 1 Report
Comments and Suggestions for Authors
The most important thing to clarify are the 127 isolates with the untyped genus.
Comments on the Quality of English LanguageNo comments about the english
Author Response
Comment 1: The most important thing to clarify are the 127 isolates with the untyped genus.
Reply: Thank you very much for the comment. For the 127 untyped isolates, they were detected through the discovery of mucor including hyphae via biopsy, autopsy or other samples like suptum. These methods cannot identify pathogens on the species level. Therefore, we have no way of knowing the specific species.
Reviewer 2 Report
Comments and Suggestions for Authors
Thank you for reporting this interesting case and reviewing the literature since the previous paper by Roden et al. I have the following questions/comments.
What were the risk factors for mucormycosis in this patient? The history of AML on treatment with azacitidine does not explain the fungal infection, unless the patient was neutropenic or on blast crisis. You should provide the leucocyte differential with absolute neutrophil and blast count, and not only the total white cell count.
You should provide some information about the mNGS result (i.e., sensitivity and specificity. Did it show only Rhizopus? Why didn't it show Staphylococcus aureus? You acknowledge negative cultures, and a nondiagnostic biopsy (which excluded malignancy but did not seem to show invasive mucormycosis; at least you don't report this). The occurrence of the infection is unusual, and the evolution is also atypical. Why was fluconazole administered for 7 days? This agent has no activity against molds.
I imagine when you write "Linazolamide" you mean linezolid, but if this is the case the dose seems low (standard is 600 mg bid).
The evolution is difficult to understand. After 101 days the lung lesion looks stable. One would argue that surgery should have been performed from the beginning. It is impossible to know what role the GM-CSF, inhaled amphotericin, topical amphotericin and thymalfasin played in this case. Overall, I am afraid the case report is very unsatisfying.
Comments on the Quality of English LanguageThe paper reads like it was written on a different language and then poorly translated. You should have it edited by either native English speaker or at least someone with working knowledge of medical English.
Author Response
Dear editor,
Thank you for your letter and those comments concerning our manuscript entitled “Oral isavuconazole combined with nebulized inhalation and bronchoscopic administration of amphotericin B for the treatment of pulmonary mucormycosis: A case report and literature review” (Manuscript ID: jof-2942954). Those comments are valuable for revising and improving our paper. We have provided a comprehensive response to all comments made by the reviewers point by point in the following pages. The revised portions are marked with red in the revised manuscript.
We deeply appreciate your consideration of our manuscript, and thanks for the editors and reviewers’ constructive advice. If you have any questions, please don’t hesitate to contact me.
Looking forward to receiving your feedback soon.
Best regards,
Feng Xu, MD, PhD,
Department of Infectious Diseases
The Second Affiliated Hospital of Zhejiang University School of Medicine, Hangzhou, China.
E-mail: xufeng99@zju.edu.cn
Comment 1: What were the risk factors for mucormycosis in this patient? The history of AML on treatment with azacitidine does not explain the fungal infection, unless the patient was neutropenic or on blast crisis. You should provide the leucocyte differential with absolute neutrophil and blast count, and not only the total white cell count.
Reply: Thank you very much for the comment. We reviewed the entire case and took the medical history again. The patient was discharged from another hospital just one day before coming to our hospital. In that hospital, he was treated for more than one month due to infection of COVID-19. It has been widely accepted that immunosuppression (either innate or acquired) plays an important role in infection of mucormycosis, including use of immunosuppressant drugs, hematologic malignancies (particularly those with prolonged neutropenia), diabetes and so on [1]. In recent years, the emergence of the Coronavirus Disease (COVID-19) disease had been associated with reports of fungal infections such as aspergillosis and mucormycosis especially among critical patients treated with steroids [2]. In the case, we think the infection of COVID-19 and the use of steroids could be the most significant risk factor. Besides, the routine blood test also showed the level of white cell and neutrophil kept low even when the C-reactive protein kept high (200mg/L). The test of T cell CD molecule showed a reduced Lymphocyte percentage (CD45+) of 3.99% (reference value: 15.00-40.00%). All of these suggested that the patient was in an immunodeficient state. We added and adjusted the detailed description of count of neutrophil and the test for T cell CD molecules (see Page 2, Line 82, 90-91).
[1] Steinbrink JM, Miceli MH. Mucormycosis. Infect Dis Clin North Am. 2021;35(2):435-452. doi:10.1016/j.idc.2021.03.009
[2] Al-Tawfiq JA, Alhumaid S, Alshukairi AN, et al. COVID-19 and mucormycosis superinfection: the perfect storm. Infection. 2021;49(5):833-853. doi:10.1007/s15010-021-01670-1
Comment 2: You should provide some information about the mNGS result (i.e., sensitivity and specificity. Did it show only Rhizopus? Why didn't it show Staphylococcus aureus? You acknowledge negative cultures, and a nondiagnostic biopsy (which excluded malignancy but did not seem to show invasive mucormycosis; at least you don't report this). The occurrence of the infection is unusual, and the evolution is also atypical. Why was fluconazole administered for 7 days? This agent has no activity against molds.
Reply: Thank you very much for the comment. We are very sorry for not showing the details of the diagnosis and treatment. The mNGS result only showed Rhizopus. We have added more information like relative abundance, number of sequences and specificity to the manuscription. However, Staphylococcus aureus was not detected. Though the suptum culture showed methicilin-resistant Staphylococcus aureus (MRSA), we think it could be a respiratory pathogen considering that the patient was recently hospitalized for over a month and being immunodeficient state.
During the course of disease, the drainage liquid test result showed several elevated tumor markers. Therefore, we performed the biopsy to rule out the possibility of tumor. The result of biopsy showed necrotic and degenerative tissue. There was no indication of invasive mucormycosis in the sampling, and we considered it to be related to the location of the sampling.
In this case, we believed that the use of steroids for treatment during hospitalization at another hospital was an important trigger. When he came to our hospital for surgery due to the fracture, a preoperative examination revealed a lesion in his left lung. Considering the immunodeficiency and infection history of COVID-19, we doubted the fungi infection. Because the suptum culture of another hospital indicated Aspergillus, the patient was treated voriconazole empirically. However, the patient's condition did not improved after the treatment for 1 week. The NGS result of the drainage liquid indicated the Rhizopus. Therefore, amphotericin B was added, and was replaced by oral isavuconazole due to the side effects. After taking isavuconazole for 2 weeks, the patient's vital signs gradually stabilize, but he presented a episode of hemoptysis. Considering that angioinvasion is a pivotal feature of mucormycosis [1], nebulized inhalation of AmB was added, and perfusion of AmB at left lower lobe basal segment through bronchoscopy was given twice.
We added the description of the mNGS result (see Page 3, Line 113-114), the result of CT-guided puncture (see Page 3, Line 118), the reason of using fluconazole (see Page 3, Line 101-102).
[1] Corzo-León DE, Uehling JK, Ballou ER. Rhizopus arrhizus. Trends Microbiol. 2023;31(9):985-987. doi:10.1016/j.tim.2023.03.013
Comment 3: I imagine when you write "Linazolamide" you mean linezolid, but if this is the case the dose seems low (standard is 600 mg bid).
Reply: Thank you very much for your correction. We have made the correction of “Linazolamide” to “linezolid” and changed the wrong dose (see Page 3, Line 121 and 126; Page 4, Line 157).
Comment 4: The evolution is difficult to understand. After 101 days the lung lesion looks stable. One would argue that surgery should have been performed from the beginning. It is impossible to know what role the GM-CSF, inhaled amphotericin, topical amphotericin and thymalfasin played in this case. Overall, I am afraid the case report is very unsatisfying.
Reply: Thank you very much for your comments. We have made corresponding modifications in the article to make the entire process clearer. We also added the chest CT image at 69 days after discharge to the article (see Page 4, Figure 2), which showed that the lesion had obvious absorption compared to the CT image of day 101 after admission.
When we got the result of mNGS, we consulted with the thoracic surgeon for the possibility of surgery. However, they denied the surgery for the immune deficiency of the patient. In addition, the patient had just experienced a femoral fracture and was in poor overall condition, making it unsuitable for surgery.
Apart from the use of anti-fungus medication, prompt rectification of predisposing conditions is indispensable and urgent [1]. The granulocyte macrophage colony- stimulating factor (GM-CSF) and thymalfasin were used as supplements to improve the immune condition. The GM-CSF was also once recommended by 3rd European Conference on Infections in Leukemia (ECIL 3) [2]. On the other hand, the inhalation and perfusion of amphotericin was used to avoid the side effects brought by intravenous administration and increase the concentration of drugs in local lesions. The details above have been presented in the part of discussion in the manuscription. We have made necessary modifications to polish the case presentation (see Page 3, Line 138-139; Page 9, Line 295-299).
[1] Rocconi R, Mazzucato S, Farina C, Grandesso S. Severe invasive pulmonary zygomycosis by Rhizomucor pusillus and concomitant severe bacterial endocarditis in acute promyelocytic leukaemia. Infez Med. 2015;23(3):265-269.
[2] Skiada A, Lanternier F, Groll AH, et al. Diagnosis and treatment of mucormycosis in patients with hematological malignancies: guidelines from the 3rd European Conference on Infections in Leukemia (ECIL 3). Haematologica. 2013;98(4):492-504. doi:10.3324/haematol.2012.065110
Comment 5: Comments on the Quality of English Language
The paper reads like it was written on a different language and then poorly translated. You should have it edited by either native English speaker or at least someone with working knowledge of medical English.
Reply: Thank you very much for the comment. We sent the article to the service of English polishing provided by MDPI and got the certificate which was submitted along with the article.
Comment 1: What were the risk factors for mucormycosis in this patient? The history of AML on treatment with azacitidine does not explain the fungal infection, unless the patient was neutropenic or on blast crisis. You should provide the leucocyte differential with absolute neutrophil and blast count, and not only the total white cell count.
Reply: Thank you very much for the comment. We reviewed the entire case and took the medical history again. The patient was discharged from another hospital just one day before coming to our hospital. In that hospital, he was treated for more than one month due to infection of COVID-19. It has been widely accepted that immunosuppression (either innate or acquired) plays an important role in infection of mucormycosis, including use of immunosuppressant drugs, hematologic malignancies (particularly those with prolonged neutropenia), diabetes and so on [1]. In recent years, the emergence of the Coronavirus Disease (COVID-19) disease had been associated with reports of fungal infections such as aspergillosis and mucormycosis especially among critical patients treated with steroids [2]. In the case, we think the infection of COVID-19 and the use of steroids could be the most significant risk factor. Besides, the routine blood test also showed the level of white cell and neutrophil kept low even when the C-reactive protein kept high (200mg/L). The test of T cell CD molecule showed a reduced Lymphocyte percentage (CD45+) of 3.99% (reference value: 15.00-40.00%). All of these suggested that the patient was in an immunodeficient state. We added and adjusted the detailed description of count of neutrophil and the test for T cell CD molecules (see Page 2, Line 82, 90-91).
[1] Steinbrink JM, Miceli MH. Mucormycosis. Infect Dis Clin North Am. 2021;35(2):435-452. doi:10.1016/j.idc.2021.03.009
[2] Al-Tawfiq JA, Alhumaid S, Alshukairi AN, et al. COVID-19 and mucormycosis superinfection: the perfect storm. Infection. 2021;49(5):833-853. doi:10.1007/s15010-021-01670-1
Comment 2: You should provide some information about the mNGS result (i.e., sensitivity and specificity. Did it show only Rhizopus? Why didn't it show Staphylococcus aureus? You acknowledge negative cultures, and a nondiagnostic biopsy (which excluded malignancy but did not seem to show invasive mucormycosis; at least you don't report this). The occurrence of the infection is unusual, and the evolution is also atypical. Why was fluconazole administered for 7 days? This agent has no activity against molds.
Reply: Thank you very much for the comment. We are very sorry for not showing the details of the diagnosis and treatment. The mNGS result only showed Rhizopus. We have added more information like relative abundance, number of sequences and specificity to the manuscription. However, Staphylococcus aureus was not detected. Though the suptum culture showed methicilin-resistant Staphylococcus aureus (MRSA), we think it could be a respiratory pathogen considering that the patient was recently hospitalized for over a month and being immunodeficient state.
During the course of disease, the drainage liquid test result showed several elevated tumor markers. Therefore, we performed the biopsy to rule out the possibility of tumor. The result of biopsy showed necrotic and degenerative tissue. There was no indication of invasive mucormycosis in the sampling, and we considered it to be related to the location of the sampling.
In this case, we believed that the use of steroids for treatment during hospitalization at another hospital was an important trigger. When he came to our hospital for surgery due to the fracture, a preoperative examination revealed a lesion in his left lung. Considering the immunodeficiency and infection history of COVID-19, we doubted the fungi infection. Because the suptum culture of another hospital indicated Aspergillus, the patient was treated voriconazole empirically. However, the patient's condition did not improved after the treatment for 1 week. The NGS result of the drainage liquid indicated the Rhizopus. Therefore, amphotericin B was added, and was replaced by oral isavuconazole due to the side effects. After taking isavuconazole for 2 weeks, the patient's vital signs gradually stabilize, but he presented a episode of hemoptysis. Considering that angioinvasion is a pivotal feature of mucormycosis [1], nebulized inhalation of AmB was added, and perfusion of AmB at left lower lobe basal segment through bronchoscopy was given twice.
We added the description of the mNGS result (see Page 3, Line 113-114), the result of CT-guided puncture (see Page 3, Line 118), the reason of using fluconazole (see Page 3, Line 101-102).
[1] Corzo-León DE, Uehling JK, Ballou ER. Rhizopus arrhizus. Trends Microbiol. 2023;31(9):985-987. doi:10.1016/j.tim.2023.03.013
Comment 3: I imagine when you write "Linazolamide" you mean linezolid, but if this is the case the dose seems low (standard is 600 mg bid).
Reply: Thank you very much for your correction. We have made the correction of “Linazolamide” to “linezolid” and changed the wrong dose (see Page 3, Line 121 and 126; Page 4, Line 157).
Comment 4: The evolution is difficult to understand. After 101 days the lung lesion looks stable. One would argue that surgery should have been performed from the beginning. It is impossible to know what role the GM-CSF, inhaled amphotericin, topical amphotericin and thymalfasin played in this case. Overall, I am afraid the case report is very unsatisfying.
Reply: Thank you very much for your comments. We have made corresponding modifications in the article to make the entire process clearer. We also added the chest CT image at 69 days after discharge to the article (see Page 4, Figure 2), which showed that the lesion had obvious absorption compared to the CT image of day 101 after admission.
When we got the result of mNGS, we consulted with the thoracic surgeon for the possibility of surgery. However, they denied the surgery for the immune deficiency of the patient. In addition, the patient had just experienced a femoral fracture and was in poor overall condition, making it unsuitable for surgery.
Apart from the use of anti-fungus medication, prompt rectification of predisposing conditions is indispensable and urgent [1]. The granulocyte macrophage colony- stimulating factor (GM-CSF) and thymalfasin were used as supplements to improve the immune condition. The GM-CSF was also once recommended by 3rd European Conference on Infections in Leukemia (ECIL 3) [2]. On the other hand, the inhalation and perfusion of amphotericin was used to avoid the side effects brought by intravenous administration and increase the concentration of drugs in local lesions. The details above have been presented in the part of discussion in the manuscription. We have made necessary modifications to polish the case presentation (see Page 3, Line 138-139; Page 9, Line 295-299).
[1] Rocconi R, Mazzucato S, Farina C, Grandesso S. Severe invasive pulmonary zygomycosis by Rhizomucor pusillus and concomitant severe bacterial endocarditis in acute promyelocytic leukaemia. Infez Med. 2015;23(3):265-269.
[2] Skiada A, Lanternier F, Groll AH, et al. Diagnosis and treatment of mucormycosis in patients with hematological malignancies: guidelines from the 3rd European Conference on Infections in Leukemia (ECIL 3). Haematologica. 2013;98(4):492-504. doi:10.3324/haematol.2012.065110
Comment 5: Comments on the Quality of English Language
The paper reads like it was written on a different language and then poorly translated. You should have it edited by either native English speaker or at least someone with working knowledge of medical English.
Reply: Thank you very much for the comment. We sent the article to the service of English polishing provided by MDPI and got the certificate which was submitted along with the article.

Reviewer 3 Report
Comments and Suggestions for Authors
The review is informative but could also be expanded to include a much more comprehensive discussion on the possible evolution of treatment strategies for pulmonary mucormycosis over time. How does the proposed combination therapy compare in efficacy, safety, and patient outcome to previously reported treatments?
Also, while the case report provided a history of the patients treatment, some additional information on the reason for choising the combination therapy should be added. To be specific, how the combination of oral isavuconazole and nebulized amphotericin B were chosen and if there were any alternative treatment options considered at that time?
What is more, the authors claimed that the treatment strategy is promising basing only on a single case. It is necessary to discuss the potential limitations of such 'generalizing' the success of this approach and see the need for more studies to validate the efficacy and safety of the combination therapy.
Minor issues:
Line 11: "pulmonary mucormycosis".
Line 18: "amphotericin B".
Throughout the whole manuscript:
1. the misspelling of "amphotericin B", but also its abbreviation (AmB) should be standardized.
2. ensure that all species names (e.g. Rhizopus microsporus) are italicized.
3. There are many instances where there is no space between a number and the unit. Those should be carefully reviewed and corrected.
Author Response
Dear editor,
Thank you for your letter and those comments concerning our manuscript entitled “Oral isavuconazole combined with nebulized inhalation and bronchoscopic administration of amphotericin B for the treatment of pulmonary mucormycosis: A case report and literature review” (Manuscript ID: jof-2942954). Those comments are valuable for revising and improving our paper. We have provided a comprehensive response to all comments made by the reviewers point by point in the following pages. The revised portions are marked with red in the revised manuscript.
We deeply appreciate your consideration of our manuscript, and thanks for the editors and reviewers’ constructive advice. If you have any questions, please don’t hesitate to contact me.
Looking forward to receiving your feedback soon.
Best regards,
Feng Xu, MD, PhD,
Department of Infectious Diseases
The Second Affiliated Hospital of Zhejiang University School of Medicine, Hangzhou, China.
E-mail: xufeng99@zju.edu.cn
Comment 1: The review is informative but could also be expanded to include a much more comprehensive discussion on the possible evolution of treatment strategies for pulmonary mucormycosis over time. How does the proposed combination therapy compare in efficacy, safety, and patient outcome to previously reported treatments?
Reply: Thank you very much for the comment. We think it is a great idea and it has already been added to the article (see Page 9, Line 268-275). We also add some previously reported trials (see Page 10, Line 306-309).
Comment 2: Also, while the case report provided a history of the patients treatment, some additional information on the reason for choising the combination therapy should be added. To be specific, how the combination of oral isavuconazole and nebulized amphotericin B were chosen and if there were any alternative treatment options considered at that time?
Reply: Thank you very much for your suggestion. Please see our reply to reviewer 2, above. We worried that the probable progression of mucormycosis for its feature of angioinvasion, so we added nebulized inhalation of AmB and perfusion of AmB (10mg once) at left lower lobe basal segment through bronchoscopy. On one hand, it can increase the concentration of drugs in local lesions and reduce the side effects brought by intravenous use. This choice was also an attempt to see if the patient could tolerate treatment. When the patient was not sensitive to medication treatment, surgical treatment needed to be considered if applicable.
Comment 3: What is more, the authors claimed that the treatment strategy is promising basing only on a single case. It is necessary to discuss the potential limitations of such 'generalizing' the success of this approach and see the need for more studies to validate the efficacy and safety of the combination therapy.
Reply: Thank you very much for your suggestion. We added a discussion on the limitations and prospect of our combination treatment at the end of the discussion section (see Page 10, Line 315-316).
Comment 4:
Minor issues:
Line 11: "pulmonary mucormycosis".
Line 18: "amphotericin B".
Throughout the whole manuscript:
1. the misspelling of "amphotericin B", but also its abbreviation (AmB) should be standardized.
2. ensure that all species names (e.g. Rhizopus microsporus) are italicized.
3. There are many instances where there is no space between a number and the unit. Those should be carefully reviewed and corrected.
Reply: Thank you very much for the comment. We have sent this article for polishing and corrected these issues in the article.
Round 2
Reviewer 2 Report
Comments and Suggestions for Authors
Why do you give more importance to the NGS results supporting mucormycosis than to the culture showing Aspergillus?
Comments on the Quality of English LanguageThank you for improving the manuscript
Author Response
Response to Reviewer 2
Comment 1: Why do you give more importance to the NGS results supporting mucormycosis than to the culture showing Aspergillus?
Reply: Thank you very much for the comment. First of all, the positive culture results of aspergillus were suggested by the outside hospital, and the sputum culture results of our hospital (see Page 3, Line 109-110) were negative. Secondly, the results of sputum culture are not accurate, because the sample of sputum is often contaminated by microorganisms in the oral cavity. Finally, we tried to treat the patient with voriconazole, but the patient's symptoms did not improve after 1 week (see Page 3, Line 102-103). Combined with the fact that the patient had been hospitalized in another hospital for 1 month, we concluded that Aspergillus was a colonizing fungi. Besides, the sample of NGS was obtained from pulmonary infection sites directly. The results of NGS had the significance of guiding treatment. Indeed, the condition of the patient was improved after anti-Rhizopus treatment.
Reviewer 3 Report
Comments and Suggestions for Authors
I am satisfied with all the answers provided.
Author Response
Comment 1: I am satisfied with all the answers provided.
Reply: Thank you very much for the comment.